# Holonomic implementation of CNOT gate on topological Majorana qubits

A. Calzona[1*], N. P. Bauer[1], B. Trauzettel[1,2]

**1** Institute of Theoretical Physics and Astrophysics, University of Würzburg,
97074 Würzburg, Germany
**2** Würzburg-Dresden Cluster of Excellence ct.qmat, Germany
* alessio.calzona@physik.uni-wuerzburg.de

December 15, 2020

## Abstract

**The CNOT gate is a two-qubit gate which is essential for universal quantum computation. A well-established approach to implement it within Majorana-based qubits relies on subsequent measurement of (joint) Majorana parities. We propose an alternative scheme which operates a protected CNOT gate via the holonomic control of a handful of system parameters, without requiring any measurement. We show how the adiabatic tuning of pair-wise couplings between Majoranas can robustly lead to the full entanglement of two qubits, insensitive with respect to small variations in the control of the parameters.**

# 1 Introduction

Topological quantum computation (TQC) is an approach to quantum computing that aims at minimizing decoherence at the hardware level, by exploiting topological properties of non-local degrees of freedom composed of non-Abelian anyons [1–3]. The latter are exotic quasiparticle excitations, which feature non-trivial exchange statistics, described by multidimensional representations of the braid group. A collection of non-Abelian anyons is embedded in a degenerate ground state manifold, which allows to non-locally store quantum information and to process it by implementing unitary transformations via braiding.

Among all non-Abelian anyons, Majorana zero-energy modes (MZMs) are the most promising ones for the development of TQC [4–8], as they are the most feasible ones in condensed matter systems. Over the last decade, seminal experiments have indeed provided strong evidence for their existence in several different platforms, such as proximitized semiconducting nanowires [9–12], chains of magnetic adatoms [13, 14], vortices within topological superconductors [15, 16], planar Josephson junctions [17, 18] and proximitized quantum spin Hall edges [19, 20].

The building block of Majorana-based TQC is the Majorana qubit, consisting of four MZMs. By physically braiding those MZMs it is possible to implement all single-qubit Clifford gates [21–23]. Those gates are topologically protected, as their outcome exclusively depends on the topology of the trajectories adiabatically followed by the anyons in a $2+1$ dimensional space. Importantly, the braiding of a single pair of MZMs can be realized in several ways, which are all equivalent to a physical exchange of the two non-Abelian anyons [24–30]. Indeed, by considering the presence of additional (hybridized) ancilla Majoranas, we can perform braiding by properly tuning pair-wise couplings between different MZMs [31, 32], or by performing sequential projective parity measurements [8,33–38]. Non-Clifford operations such as the T gate cannot be realized via Majorana braiding and necessarily rely on implementations that are not topologically protected and require additional error correction schemes such as magic state distillation [23, 39].

To achieve universal quantum computation, single-qubit gates must be supplemented with an entangling gate, such as the CNOT gate. Unfortunately, this two-qubit Clifford gate cannot be realized within a scalable architecture by exclusively using Majorana braiding operations [22, 40]. The measurement-based approach allows us to overcome this issue by implementing the CNOT gate by performing high-fidelity projective measurements of the (joint) Majorana parities [8, 35, 41–44]. However, while measurement-based TQC has proven to be extremely valuable for the future development of a fully scalable topological quantum computer, the required measurement protocols still represent a formidable challenge [35, 45, 46]. For the time being, it is therefore desirable to devise and characterize alternative schemes, which do not rely on high-fidelity measurements but still allow to robustly entangle distinct topological qubits.

In this work, we propose a measurement-free realization of the CNOT gate based on a holonomic approach. The key idea of holonomic quantum computation is to exploit non-

Abelian geometrical phases to implement unitary operations on a degenerate eigenspace of the underlying Hamiltonian [47]. Those gauge-invariant phases emerge when the parameters of the system are tuned along degeneracy-preserving closed loops in parameter space. This approach is rather general and has been successfully exploited in non-topological quantum computation schemes [47–49]. Therefore, it is interesting to utilize holonomic techniques also in TQC. Indeed, the braiding of Majoranas itself can be interpreted as a holonomic process, where the system follows specific, topologically-protected loops in the three-dimensional parameter space of pair-wise Majorana couplings [8, 31]. The advantage of the holonomic description of the braiding is that it can be easily generalized, both by considering loops with a different structure, within the same parameter space, and/or by considering a different parameter space altogether. In the first case, a careful modification of the loops can effectively implement non-Clifford (and non-topological) gates, such as the T gate [50]. We consider the second case and demonstrate that it is possible to implement entangling gates, such as the CNOT gate, by working in specific parameter spaces of a two-qubit system. In presence of physical constraints on the fermion parity of individual qubits, provided, for instance, by a finite charging energy [30, 32, 36], our holonomic entangling scheme is robust with respect to the presence of otherwise detrimental couplings between MZMs and/or limited control of system parameters.

The article is organized as follows. In Sec. 2, we introduce the structure of the specific Majorana qubits under consideration. We also briefly review the concepts of holonomic quantum computation. In Sec. 3, we propose two different holonomic implementations of two-qubit entangling gates, whose robustness is analyzed in Sec. 4. Finally, we summarize and discuss our findings in Sec. 5.

## 2 Model

For the sake of generality, we do not focus on a specific physical implementation of Majorana qubits. Instead, we discuss a generic low-energy effective model consisting of several couples of Majorana modes (or Majoranas). The latter are described by the self-adjoint operators $\gamma_j = \gamma_j^\dagger$ which obey

$$\{\gamma_j, \gamma_k\} = 2\delta_{j,k}. \tag{1}$$

If they commute with the system Hamiltonian $[H, \gamma_j] = 0$, we refer to them as MZMs. Two Majorana operators define a (possibly non-local) fermion

$$f_{jk} = \frac{1}{2}\left(\gamma_j - i\gamma_k\right), \tag{2}$$

whose associated parity operator reads

$$P_{jk} = i\gamma_j\gamma_k = 1 - 2f_{jk}^\dagger f_{jk}. \tag{3}$$

Two degenerate states with opposite parity $P_{jk}|0_{jk}\rangle = |0_{jk}\rangle$ and $P_{jk}|1_{jk}\rangle = -|1_{jk}\rangle$ can be therefore associated with every pair of MZMs. As the global fermion parity of an isolated system is fixed, a working Majorana qubit requires the presence of four different MZMs $\gamma_i$ (with $i = 1, 2, 3, 4$) [22]. Without loss of generality, we consider the global parity of the

Majorana qubit to be even and choose the following basis for the computational space of the qubit

$$
\begin{aligned}
|\tilde{0}\rangle &= |0_{12}0_{34}\rangle, \\
|\tilde{1}\rangle &= |1_{12}1_{34}\rangle.
\end{aligned}
\tag{4}
$$

Our goal is to operate on the Majorana qubit by following an holonomic approach, i.e. by changing in time some of its parameters. However, a system consisting only of four MZMs does not allow for such a manipulation, as it does not feature tunable parameters. To overcome this issue, we add two additional Majoranas, $\gamma_0$ and $\gamma_5$, to the system and consider the general Hamiltonian

$$
H = -\sum_{j=1}^{5} c_j P_{0j},
\tag{5}
$$

which describes pair-wise couplings within a "five-point star" scheme, as sketched in Fig. 1(a). When no operations are performed on the qubit, it is in the idle configuration which features a single non-vanishing coupling strength $c_5 = \Theta > 0$ and leaves us with the four MZMs $\gamma_i$ ($i = 1, 2, 3, 4$). Without loss of generality, we consider the total fermion parity of the system to be even $\mathcal{P} = P_{12}P_{34}P_{05} = 1$. In this case, the computational space is spanned by the two degenerate ground states

$$
\begin{aligned}
|0\rangle &= |\tilde{0}\rangle|0_{05}\rangle = |0_{12}0_{34}0_{05}\rangle \\
|1\rangle &= |\tilde{1}\rangle|0_{05}\rangle = |1_{12}1_{34}0_{05}\rangle.
\end{aligned}
\tag{6}
$$

The excited states, with the same total fermion parity, are separated from the ground states by an energy gap $\Delta E = 2\Theta$.

The advantage of this five-point star architecture is that it is tunable. Indeed, it is possible to tune the coupling strengths $c_j$ away from the idle configuration without destroying the qubit, keeping the fixed-parity computational space degenerate and separated from the excited states by a finite energy gap. A sufficient condition for the stability of the qubit is that, at each time $t$, either one or two of the five different couplings strengths $c_j$ must be non-vanishing [31, 50, 51].

## 2.1 Holonomic description of Majorana braiding

We now briefly review a protocol which allows us to braid a couple of MZMs, say $\gamma_1$ and $\gamma_2$, thus implementing a topological quantum gate. This protocol has been extensively discussed in Ref. [8, 31, 50] To this end, we focus on the Hamitonian

$$
H_{1\leftrightarrow 2}(t) = -\chi_x(t)P_{02} - \chi_y P_{01} - \chi_z(t)P_{05}.
\tag{7}
$$

Starting from the idle configuration at $t = \tau_0 = 0$, we tune the three coupling strengths $\chi_j(t)$ (with $j = x, y, z$) along the closed loop $\Gamma$ in the three-dimensional parameter space shown Fig. 1(b). The loop consists of six straight lines connecting six *key* configurations reached at times $t = \tau_k$ (with $k = 0, \dots, 6$), see Tab. 1 for more details. At the end of the protocol $t = T = \tau_6$, the system goes back to the original idle configuration, i.e. $\chi_j(\tau_6) = \chi_j(\tau_0)$. In between two subsequent configurations, two parameters are kept fixed while the third

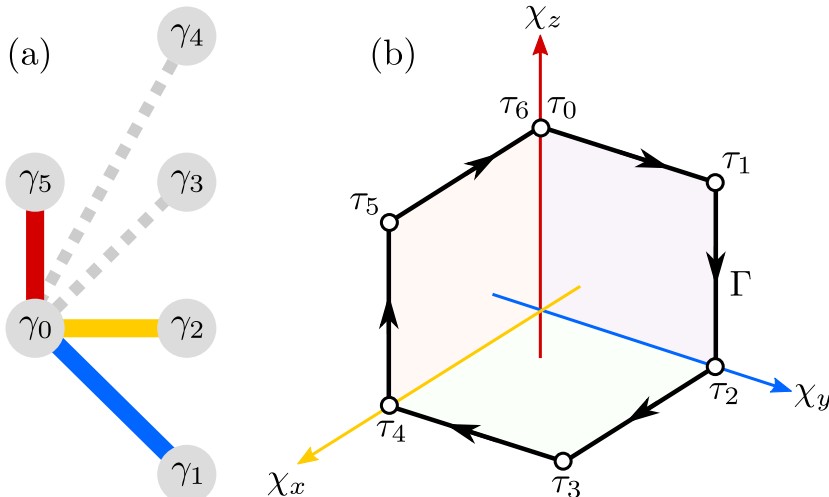

Figure 1: (a) Sketch of a single Majorana qubit consisting of six MZMs. They are connected by five lines representing the five pair-wise couplings $P_{0j}$ which enter the qubit Hamiltonian in Eq. (5). In the idle configuration, only Majoranas $\gamma_0$ and $\gamma_5$ are coupled (thick red line), leaving the other four Majoranas at zero energy. The holonomic braiding of MZMs $\gamma_1$ and $\gamma_2$ requires the additional manipulation of couplings $P_{01}$ and $P_{02}$ (blue and yellow line, respectively). (b) The closed loop $\Gamma(t)$ followed by the three coupling strengths $\chi_x$, $\chi_y$ and $\chi_z$ specified in Eq. (7) during the clockwise braiding of MZMs $\gamma_1$ and $\gamma_2$. The "anchor points" $\Gamma(\tau_j)$, which are listed in Tab. 1, are highlighted with small bullets.

one smoothly interpolates between 0 and its maximum value (or viceversa). For the sake of concreteness, in what follows, we consider the interpolating function

$$g(l) = \frac{1 - \cos(\pi l)}{2} \tag{8}$$

with $g(0) = 0$ and $g(1) = 1$.

This protocol clearly satisfies the (above-mentioned) sufficient condition which ensures the preservation of the qubit computational space. We can therefore study the adiabatic time evolution of a generic initial state $|i\rangle = \alpha|0\rangle + \beta|1\rangle$. Once the system is back in the idle configuration, at time $t = T$, the final state $|f\rangle$ must be related to the initial one by a $U(2)$ transformation $|f(T)\rangle = \mathcal{U}_\Gamma|i\rangle$. Because of the degeneracy of the computational space, the dynamical phase picked by $|0\rangle$ during the time evolution is the same as the one picked by $|1\rangle$. Therefore, a non-trivial $\mathcal{U}_\Gamma$ can only emerge as a non-Abelian Berry phase, which depends on the geometrical properties of the loop $\Gamma$. Notably, it is independent of the specific values of $\tau_j$ and the interpolating function $g$. Moreover, the conservation of $[P_{34}, H_{1\leftrightarrow2}(t)] = 0$ and $[\mathcal{P}, H_{1\leftrightarrow2}(t)] = 0$ implies that $\mathcal{U}_\Gamma$ is diagonal in the basis $\{|0\rangle, |1\rangle\}$. The Berry phase picked by the state $|0\rangle$ ($|1\rangle$) equals (minus) the solid angle $\Omega_\Gamma$ enclosed by the loop $\Gamma$ [8,31,50]. For the loop depicted in Fig. 1(b), up to an overall phase, this corresponds to the unitary matrix

$$\mathcal{U}_\Gamma = \begin{pmatrix} 1 & 0 \\ 0 & -i \end{pmatrix}. \tag{9}$$

Therefore, the holonomic scheme we briefly reviewed corresponds to a quantum phase gate (also known as $\pi/4$ gate).

| $t$ | $\chi_x(t)$ | $\chi_y(t)$ | $\chi_z(t)$ |
|---|---|---|---|
| $\tau_0 = 0$ | 0 | 0 | $\Theta$ |
| $\tau_1$ | 0 | $\Theta_y$ | $\Theta$ |
| $\tau_2$ | 0 | $\Theta_y$ | 0 |
| $\tau_3$ | $\Theta_x$ | $\Theta_y$ | 0 |
| $\tau_4$ | $\Theta_x$ | 0 | 0 |
| $\tau_5$ | $\Theta_x$ | 0 | $\Theta$ |
| $\tau_6 = T$ | 0 | 0 | $\Theta$ |

Table 1: Values of the coupling strengths $\chi_i(t)$ at the *key* configurations $t = \tau_\alpha$ (with $\alpha = 0, \ldots, 6$) along the closed loop shown in Fig. 1(b).

This holonomic protocol is completely equivalent to the physical clockwise braiding of $\gamma_1$ and $\gamma_2$. This can be easily understood by tracking the motion of the zero-energy modes, initially associated with $\gamma_1$ and $\gamma_2$, throughout the evolution of the system along $\Gamma$ [31,51]. It is therefore not surprising that this holonomic protocol inherits the same topological protection featured by the physical braiding of MZMs. A single topological MZM can be completely decoupled from other Majoranas with exponential accuracy, for instance, by varying its distance from the other Majoranas. This has two important consequences: (i) It guarantees the exponential protection of the degeneracy of the computational space, thus preventing the onset of unwanted dynamical phase differences. (ii) It exponentially confines the loop $\Gamma$ on the three coordinate planes $\chi_j = 0$. Therefore, the enclosed solid angle is always $\Omega_\Gamma = \pi/2$, regardless of deviations from the ideal "cubic" shape of the loop $\Gamma$ in Fig. 1(b) which may arise due to a limited control on the non-vanishing couplings $\chi_k$ [8,31,50]. Hence, the unitary operation implemented by the holonomic protocol can approach $\mathcal{U}_\Gamma$ in Eq. (9) to exponential accuracy.

With respect to the physical braiding, the advantage of the holonomic approach is that it can be easily generalized to different classes of loops and/or to different parameter spaces. In the first case, for example, by halving the solid angle enclosed by the loop $\Gamma$, it is possible to implement a T gate (also known as $\pi/8$ gate). While the full topological protection is lost, this implementation of the T gate can still take advantage of geometrical protection. Indeed, perturbations on the loop which do not change the enclosed solid angle $\Omega$ do not affect the gate outcome [50]. In the next section, we explain how two Majorana qubits can be entangled by holonomic schemes.

## 3 Entangling gates

We consider a two-qubit system, consisting of two copies of the five-point star architecture as shown in Fig. 2(a). It features twelve different Majorana modes $\gamma_j^\alpha$, with $j = 0, \ldots 5$ and the qubit index $\alpha = A, B$. In complete analogy with the single-qubit case, we define inter-coupling pairings $P_{jk}^\alpha = i\gamma_j^\alpha \gamma_k^\alpha$ and introduce the idle configuration, which features two non-vanishing couplings $H_{idle} = -P_{05}^A - P_{05}^B$ [see Fig. 2(a)]. Moreover, without loss of generality, we assume each qubit to obey a global even fermion parity, i.e. $\mathcal{P}^\alpha = -i\gamma_1^\alpha \gamma_2^\alpha \gamma_3^\alpha \gamma_4^\alpha \gamma_0^\alpha \gamma_5^\alpha = 1$. The computational space of the whole system in the idle configuration is thus spanned by four

degenerate ground states

$$\{|0_A0_B\rangle, |0_A1_B\rangle, |1_A0_B\rangle, |1_A1_B\rangle\}, \tag{10}$$

where the single-qubit even-parity states $|0_\alpha\rangle$ and $|1_\alpha\rangle$ are defined as in Eq. (6).

While holonomic single-qubit gates can be easily implemented by manipulating the intra-qubit couplings $P_{ij}^A$, two-qubit gates require the manipulation of terms in the Hamiltonian which act on both qubits, such as the inter-qubit couplings $\mathcal{I}_{jk} = i\gamma_i^A\gamma_j^B$. Importantly, these terms do not commute with the fermion parity of each qubit, i.e. $[\mathcal{P}^\alpha, \mathcal{I}_{jk}] \neq 0$. Therefore, they can potentially push the two-qubit system out of its four-dimensional computational space, spanned by the even-even states in Eq. (10). This is precisely what happens if we braid Majoranas $\gamma_1^A$ and $\gamma_2^B$, for example by holonomically tuning parameters $P_{05}^B$, $P_{02}^B$, and $\mathcal{I}_{10}$. Such an operation would entangle the two subsystems $A$ and $B$ but it would also induce leakages out of the two-qubit computational space. This specific example is a consequence of the more general non-entangling rule, introduced by Bravyi in 2006 [22] stating that no entangling gates between distinct qubits can be realized just by braiding Ising anyons. For the sake of completeness, we acknowledge that it is possible to entangle a couple of Majorana qubits, defined in a peculiar way such that they share a pair of Ising anyons, just by using Majorana braiding [52]. However, this approach is not scalable since the image of the braiding group is a nontrivial subgroup of the Clifford group for $n \geq 3$ qubits, meaning that there exist Clifford gates which cannot be realized only by braiding [40].

To overcome the limitations posed by the non-entangling rule, it is necessary to devise novel holonomic protocols that go beyond the simple braiding of MZMs and preserve the fermion parity $\mathcal{P}^\alpha$ of each individual qubit. To this end, it is possible to follow two different approaches. One possibility is to manipulate more complicated operators that act on both qubits while still commuting with each $\mathcal{P}^\alpha$. Alternatively, it is possible to ensure the conservation of fermion parity by external means, for instance, by making unfavorable single-electron tunnelings between the qubits. In what follows, we propose two protocols which are based on the first and second approach, respectively.

## 3.1 The $4\gamma$ protocol

To devise an holonomic procedure which fully entangles the two qubits, we introduce the operator

$$O^{4\gamma} \equiv P_{12}^A P_{01}^B = -\gamma_1^A \gamma_2^A \gamma_0^B \gamma_1^B = \mathcal{I}_{20}\mathcal{I}_{11}. \tag{11}$$

It represents an interacting term which involves the two qubits but preserves their individual parities $[O^{4\gamma}, \mathcal{P}^\alpha] = 0$. We argue that it allows us to realize a holonomic entangling gate by considering the time-dependent Hamiltonian

$$H^{4\gamma}(t) = -\chi_x(t)P_{02}^B - \chi_y(t)P_{12}^A P_{01}^B - \chi_z(t)P_{05}^B, \tag{12}$$

where the coupling strengths $\chi_j(t)$ adiabatically follow the loop $\Gamma$ depicted in Fig. 1(b) and described in Tab. 1.

Before presenting a detailed and rigorous analysis of this protocol, it is useful to develop some intuition about its entangling capabilities. To this end, we notice that the Hamiltonian $H^{4\gamma}$ closely resembles the one which would braid Majoranas $\gamma_1^B$ and $\gamma_2^B$ within the same qubit [see Eq. (7)]. The key difference is the presence of the operator $O^{4\gamma}$, which replaces the simple intra-qubit coupling $P_{01}^B$ with the product $P_{12}^A P_{01}^B$. This allows to effectively implement

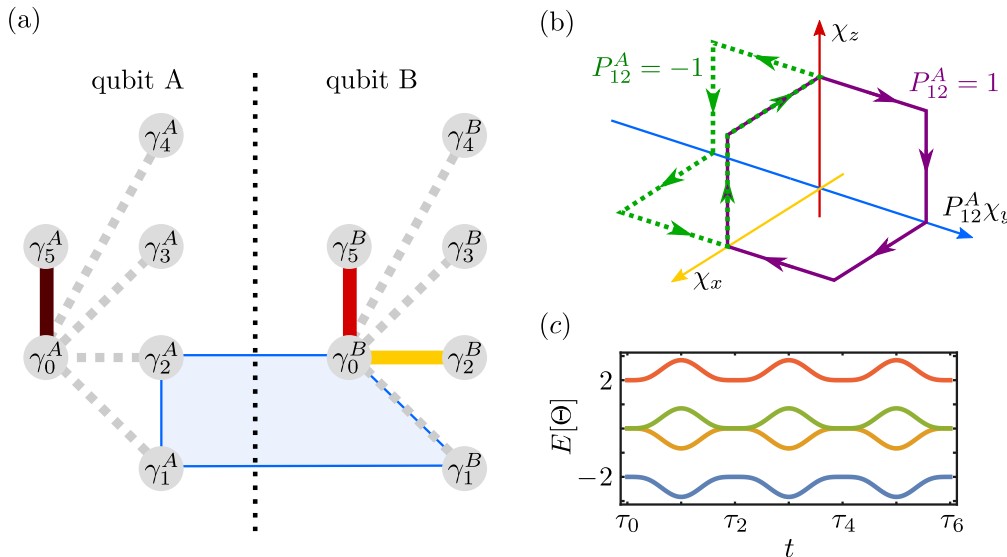

Figure 2: (a) Sketch of a two-qubit system consisting of 12 Majoranas. In the idle configuration, only the inter-qubit couplings $P_{05}^A$ and $P_{05}^B$ are non-vanishing and they are represented with dark red and red lines, respectively. The $4\gamma$ entangling protocol, described in Sec. 3.1, requires the control of the additional intra-qubit coupling $P_{02}^B$ (yellow line) and inter-qubit interaction $\mathcal{I}_{11}\mathcal{I}_{20}$ (blue area). (b) Intuitive understanding of the effect of the inter-qubit interaction term: depending on the parity of the control MZM pair $\gamma_1^A$ and $\gamma_2^A$, the holonomic protocol effectively corresponds either to a clockwise (purple loop) or anticlockwise (green dotted loop) effective braiding of the target Majoranas $\gamma_1^B$ and $\gamma_2^B$. (c) Spectrum of the Hamiltonian $H^{4\gamma}(t)$ along the closed loop $\Gamma$. Each line is 16-fold degenerate. Energies are in units of $\Theta_x = \Theta_x = \Theta$. We consider the six *key* configurations to be equally spaced in time, i.e. a constant $\tau_{j+1} - \tau_j$.

a *controlled braiding*. Indeed, the parity of Majoranas $\gamma_1^A$ and $\gamma_2^A$ on the control qubit $A$, controls the direction of the braiding of $\gamma_0^B$ and $\gamma_1^B$ on the target qubit $B$: for $P_{12}^A = \pm 1$, the effective braiding on qubit $B$ is either clockwise $(+1)$ or anti-clockwise $(-1)$, as shown in Fig. 2(b) with solid purple and dotted green lines, respectively. This explains the entangling capability of the procedure.

In order to fully characterize the holonomic protocol, it is necessary to study the adiabatic time evolution of the four even-even ground states in Eq. (10). Initially, at time $t = 0$, we assume the system to be in one of the states $|\psi_{nm}(t = 0)\rangle = |n_A m_B\rangle$, with $n, m \in \{0, 1\}$. Since the parity operators $P_{34}^A$ and $P_{34}^B$ are conserved throughout the whole time evolution, they allow us to conveniently label the states at every time $t$, that is

$$
\begin{aligned}
P_{34}^A|\psi_{nm}(t)\rangle &= (-1)^n|\psi_{nm}(t)\rangle, \\
P_{34}^B|\psi_{nm}(t)\rangle &= (-1)^m|\psi_{nm}(t)\rangle.
\end{aligned}
\tag{13}
$$

Importantly, the four states $|\psi_{nm}(t)\rangle$ are always degenerate. This can be proven by identifying zero energy operators which allow us to transform between these four states. For $0 \le t \le \tau_2$

(see Tab. 1 and Fig. 1), we find that

$$R_A = \gamma_2^A \gamma_3^A \gamma_0^B \gamma_5^B, \tag{14}$$

$$R_B = \gamma_2^B \gamma_3^B \tag{15}$$

represent these operators as they commute with the Hamiltonian $[H(t), R_A] = [H(t), R_B] = 0$ and anticommute with the respective conserved quantity $\{P_{34}^\alpha, R_\alpha\} = 0$. Analogously, zero energy operators can also be found for the following steps, i.e. for $\tau_2 \leq t \leq \tau_6 = T$. The energy spectrum of the system, obtained by diagonalizing the full Hamiltonian $H^{4\gamma}(t)$, confirms the degeneracy of states $|\psi_{nm}\rangle$ and the presence of a finite energy gap, which separates them from excited states [see Fig. 2(c)].

The adiabatic theorem, together with the conservation of the parities $P_{34}^\alpha$, allows us to relate the final states $|\psi_{nm}(T)\rangle$ to the initial ones via a $U(4)$ diagonal matrix $\mathcal{U}_\Gamma^{ent}$. Up to a global phase, this transformation only depends on the geometric properties of the loop $\Gamma$ in the parameter space. In order to find $\mathcal{U}_\Gamma^{ent}$, we compute the Berry curvature associated with each state (see App. A.1 for more details)

$$\begin{aligned}
\vec{F}_{nm} &= \nabla_{\vec{\chi}} \times i\langle \psi_{nm}(\vec{\chi})|\nabla_{\vec{\chi}}|\psi_{nm}(\vec{\chi})\rangle \\
&= (-1)^{m+n+1} \frac{\vec{\chi}}{2|\vec{\chi}|^3}.
\end{aligned} \tag{16}$$

Therefore, analogously to the simple holonomic braiding, the Berry phases picked up by each element of the basis in Eq. (10) only depends on the solid angle enclosed by the closed loop $\Gamma$ in parameter space. In particular, we get

$$|\psi_{nm}(T)\rangle = \exp\left[(-1)^{m+n} i\frac{\pi}{4}\right] |\psi_{nm}(t=0)\rangle \tag{17}$$

which, up to a global phase, corresponds to the unitary transformation

$$\mathcal{U}_\Gamma^{ent} = \begin{pmatrix} 1 & 0 & 0 & 0 \\ 0 & -i & 0 & 0 \\ 0 & 0 & -i & 0 \\ 0 & 0 & 0 & 1 \end{pmatrix}. \tag{18}$$

The entangling power of this gate is $\text{EP}(\mathcal{U}_\Gamma^{ent}) = 2/9$, which means that it is able to fully entangle the two qubits [53]. For instance, starting from the product state $|i\rangle = (|0_A\rangle + |1_A\rangle) \otimes (|0_B\rangle + |1_B\rangle)/2$, the holonomic procedure generates the maximally entangled state

$$\mathcal{U}_\Gamma^{ent}|i\rangle = \frac{|0_A 0_B\rangle - i|0_A 1_B\rangle - i|1_A 0_B\rangle + |1_A 1_B\rangle}{2}. \tag{19}$$

These results have also been confirmed numerically, by simulating the adiabatic time evolution of the system with the QuTip package [54, 55] [see, for example, Fig. 4(a)]. By applying additional single-qubit Hadamard (H) and phase (S) gates, which can be implemented using, for example, holonomic braiding as discussed in Sec. 2.1, it is possible to obtain the control-Z gate

$$\text{CZ} = \mathcal{U}_\Gamma^{ent\,\dagger}(S \otimes S) \tag{20}$$

as well as the CNOT gate

$$\text{CNOT} = (I \otimes H)\text{CZ}(I \otimes H). \tag{21}$$

The practical realization of this holonomic entangling protocol is necessarily characterized by a finite execution time $T$, which inevitably induces deviations from the adiabatic result in Eq. (19): a decrease in $T$ increases the probability of unwanted transitions between the computational space and the excited states. Importantly, the spectrum associated with the *controlled braiding*, which is depicted in Fig. 2(c), is completely equivalent to the one featured by a standard single-qubit holonimic braiding protocol [see, for instance, Eq. (7)]. Diabatic errors in our $4\gamma$ entangling protocol are therefore expected to be equivalent to the ones which characterize single-qubit holonomic braidings, whose properties have been already extensively studied and reviewed [56–61].

We mention in passing that the unitary operator $\mathcal{U}_\Gamma^{ent}$ can also be obtained by using a measurement-only approach. According to Ref. [34], the latter would require four subsequent projective forced measurements of the very same parity operators, which we tune in $H^{4\gamma}$. That is

$$\mathcal{U}_\Gamma^{ent} \propto \Pi_{P_{05}^B}^+ \Pi_{P_{02}^B}^+ \Pi_{P_{12}^A P_{01}^B}^+ \Pi_{P_{05}^B}^+ \tag{22}$$

where $\Pi_P^+ = (1+P)/2$ is the projector on the eigenstate of the parity operator $P$ with eigenvalues $+1$. This draws a formal equivalence between the two approaches. While our holonomic protocol does not rely on the realization of projective (joint) parity measurements, it poses (at least) two other important experimental challenges: (i) The implementation of a tunable four-Majorana interaction term $O^{4\gamma}$ and (ii) the detrimental effect of parasitic couplings. The latter correspond to unwanted pair-wise couplings, which can appear during the manipulation of the Majoranas. For example, at time $t = \tau_1$, Majorana $\gamma_0^B$ is coupled with $\gamma_5^B$ [red line in Fig. 2(a)] and it simultaneously interacts with $\gamma_2^A$, $\gamma_1^A$ and $\gamma_1^B$ [blue area in Fig. 2(a)]. In this configuration, it might be difficult to prevent the onset of additional small parasitic couplings between the involved Majoranas, such as $\mathcal{I}_{25} = i\gamma_2^A\gamma_5^B$ or $P_{12}^A = i\gamma_1^A\gamma_2^A$. Unfortunately, these terms do not conserve the parity of individual qubits and/or split the ground state degeneracy, spoiling the holonomic entangling procedure. In the next section, we propose an alternative holonomic approach that overcomes those two aforementioned drawbacks.

## 3.2 The double tunneling protocol

In this section, we take advantage of the framework developed for the study of the $4\gamma$ protocol and devise a new holonomic scheme that features high resilience against parasitic couplings and only requires to tune pair-wise Majorana couplings. As discussed before, such an approach necessarily relies on external means to preserve the parity of each individual qubit. To this end, we consider the additional time-independent Hamiltonian

$$H_\mathcal{E} = -\mathcal{E}(\mathcal{P}^A + \mathcal{P}^B) \tag{23}$$

which makes single-electron tunneling between the two qubits energetically unfavorable (in the regime of a significant magnitude of $\mathcal{E}$). Importantly, such a Hamiltonian naturally emerges in systems where each qubit consists of a floating superconducting island with a finite charging energy. This feature has previously been exploited to protect qubits from quasiparticle poisoning and to allow joint-parity measurements of more than two Majoranas [31,35,36,46].

The presence of a finite $H_\mathcal{E}$ allows us to trade the interacting term $O^{4\gamma} = \mathcal{I}_{20}\mathcal{I}_{11}$ [see Eq. (11)] with a simpler and more feasible one

$$O^{2\gamma+2\gamma} = \mathcal{I}_{20} + \mathcal{I}_{11}, \tag{24}$$

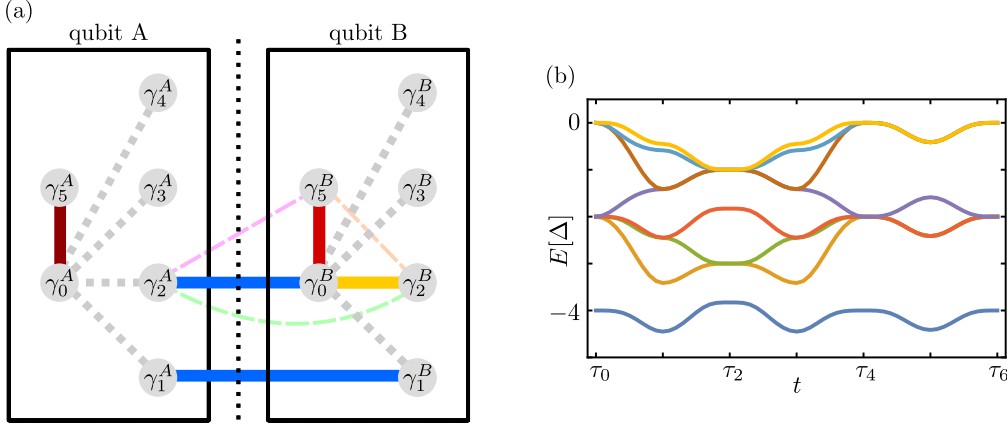

Figure 3: (a) Sketch of the two-qubit system, with constraints on the single-qubit parities imposed by a finite $H_\mathcal{E}$ (black rectangles). The intra-qubit couplings $P_{05}^A$ and $P_{05}^B$ which are non-vanishing in the idle configuration are highlighted with dark red and red lines, respectively. The double tunneling protocol requires the additional control of the intra-qubit coupling $P_{02}^B$ (yellow line) and of the inter-qubit couplings $\mathcal{I}_{20}$ and $\mathcal{I}_{11}$ (blue lines). Possible parasitic couplings $\mathcal{I}_{25}$, $\mathcal{I}_{22}$ and $P_{02}^B$ are shown with dashed purple, green and orange lines, respectively. (b) Spectrum of the Hamiltonian $H^{dt}(t)$ along the closed loop $\Gamma$. Only negative energies are shown. Each line is 4-fold degenerate. Energies are in units of $\Theta_x = \Theta_x = \mathcal{E} = \Theta$. We considered the six *key* configurations to be equally spaced in time, i.e. $\tau_{j+1} - \tau_j = T/6$.

even though the latter does not commute with the qubit parities $\mathcal{P}^\alpha$. In order to develop some intuition, we focus at first on the limit of large $\mathcal{E}$ so that the individual inter-qubit tunnelings $\mathcal{I}_{20}$ and $\mathcal{I}_{11}$ are highly suppressed. In this limit, only virtual cotunneling processes can happen such as

$$(O^{2\gamma+2\gamma})^2 = 2 + 2O^{4\gamma} \tag{25}$$

whose effect is analogous to the interacting term $O^{4\gamma}$. Hence, the time-dependent Hamiltonian

$$H^{dt}(t) = H_\mathcal{E} - \chi_x(t)P_{02}^B - \chi_y(t)\left(\mathcal{I}_{20} + \mathcal{I}_{11}\right) - \chi_z(t)P_{05}^B, \tag{26}$$

is likely to function in a similar way as previously discussed for $H^{4\gamma}$. This is confirmed by the careful analysis of the protocol $H^{dt}(t)$ which we carry out below. Importantly, we argue that it is not necessary to work in the limit of large $\mathcal{E}$, as the holonomic entangling scheme can be successfully implemented even for a finite $\mathcal{E} \sim \Theta$ (see Fig. 4). For the sake of simplicity, in Eq. (26), we have considered the same coupling strength $\chi_y(t)$ for both $\mathcal{I}_{20}$ and $\mathcal{I}_{11}$. In App. B, we show that differences in the two coupling strengths are not detrimental for the holonomic entanglement procedure.

We characterize the protocol based on $H^{dt}(t)$ along the lines of the previous section. In particular, we observe that the parity operators $P_{34}^A$ and $P_{34}^B$ are still conserved and that operators like $R_A$ and $R_B$, transforming between the states of the computational space, are still at zero energy. This guarantees the degeneracy of the four groundstates $|\psi_{nm}(t)\rangle$ throughout the whole protocol, as nicely confirmed by the spectrum plotted in Fig. 3(b) and obtained by the exact diagonalization of the full Hamiltonian $H^{dt}(t)$. Hence, after the adiabatic evolution of the system, the final states $|\psi_{nm}(T)\rangle$ are related to the initial ones $|\psi_{nm}(0)\rangle$ via a $U(4)$ diagonal matrix which, up to a global phase, depends only on the geometrical properties of

the closed loop $\Gamma$. In order to determine this unitary transformation, we compute the Berry phase picked up by each state by integrating the Berry connection along $\Gamma$ (see App. A.2). Even though the Berry curvature differs from the simple form in Eq. (16), we find that, up to a global phase, the implementation of the protocol based on Hamiltonian $H^{dt}(t)$ results in the same unitary transformation $\mathcal{U}_\Gamma^{ent}$ we obtained within the $4\gamma$ protocol. Importantly, this result does not depend on $\mathcal{E}$ as long as the latter is finite. The protocol is, therefore, able to maximally entangle the two qubits and it can be straightforwardly turned into a CNOT gate by adding single-qubit Clifford gates according to Eq. (21).

All these results have been confirmed numerically by simulating the time evolution of the system and testing the validity of Eq. (19) [see Fig. 4(a)]. Importantly, numerical simulations represent also a valuable tool to fully characterize the protocol by inspecting its robustness against non-adiabatic effects, parasitic couplings, and poor control of the tuning parameters. These aspects, which are of fundamental importance when it comes to practical implementations of the proposed entangling scheme, are carefully discussed in the next section.

# 4 Robustness of the entangling protocol

So far, we have studied the entangling protocol based on the Hamiltonian $H^{dt}(t)$ under ideal conditions. Indeed, we have considered the time evolution of the system while the three control parameters $\chi_j(t)$ are adiabatically and precisely tuned along the loop $\Gamma$. This raises the question to which degree the resulting unitary operation $\mathcal{U}_\Gamma^{ent}$ is robust with respect to non-ideal effects, which arise under more realistic conditions.

## 4.1 Adiabaticity

Strictly speaking, the existence of a finite energy gap $\Delta E$ between the ground state manifold and the excited states guarantees the applicability of the adiabatic theorem only in the limit $T \to \infty$. Protocols with a shorter time duration $T$ might indeed feature diabatic transitions (e.g. of Landau-Zener type) which push the system out of computational space and spoil the holonomic quantum gate.

In order to analyze and quantify this effect, we numerically simulate the time evolution of the initial state $|i\rangle = (|0_A\rangle + |1_A\rangle) \otimes (|0_B\rangle + |1_B\rangle)/2$ for several different durations $T$ of the whole protocol. The final states $|f(T)\rangle$ can then be compared with the expected result $|f(\infty)\rangle = \mathcal{U}_\Gamma^{ent}|i\rangle$ [see Eq. (19)] by computing the overlaps $|\langle f(\infty)|f(T)\rangle|^2$. The latter are plotted in Fig. 4(a) for $\mathcal{E} = \Theta$ and show that the time evolution of the system behaves adiabatically for $T \gtrsim 100\Delta^{-1}$. Faster implementations of the protocol would result in $|f(T)\rangle \neq |f(\infty)\rangle$. The oscillations featured by the overlap $|\langle f(\infty)|f(T)\rangle|^2$ for short $T$ can be understood as interference patterns resulting from subsequent transitions between ground and excited states, in analogy with the Landau-Zener-Stückelberg effect [51,62].

Let us define the "adiabatic threshold" $T_{ad}$ as the minimal duration of the holonomic process for which diabatic effects become negligible, say $1 - |\langle f(\infty)|f(T)\rangle|^2 < 10^{-3}$ for all $T \geq T_{ad}$. This time scale clearly depends on the spectrum of the system during the time evolution which, in turn, depends on the energy $\mathcal{E}$ and on the maximum coupling strengths $\Theta_i$ featured by the $\Gamma$ loop specified in Tab. 1. In Fig. 4(b), we numerically compute $T_{ad}$ as a function of $\mathcal{E}$ for three different values of $\Theta_y$ while keeping $\Theta_x = \Theta$ fixed. Notably, both large and small values of $\mathcal{E}$ are detrimental for the adiabaticity of the protocol: For a given

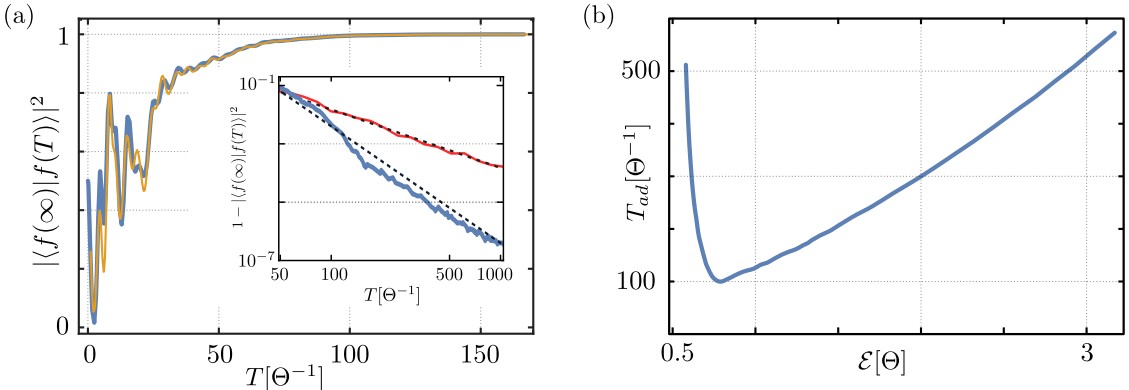

Figure 4: (a) Study of diabatic effects for a finite duration $T$ of the "double tunneling" entangling scheme. The overlap $|\langle f(\infty)|f(T)\rangle|^2$ is plotted as a function of $T$ (in units of $\Theta^{-1}$). The blue line refers to the ideal protocol, i.e. Eq. (26), while the thin orange one considers the presence of finite parasitic coupling with $V = \Theta/2$ [see Eqs. (28) and (29)]. The inset display $1 - |\langle f(\infty)|f(T)\rangle|^2$ on a logarithmic scale. The blue line refer to the same ideal protocol plotted in the main panel; the red one is obtained with a different choice of the interpolating function: $\tilde{g}(l) = l$ instead of the $g(l)$ in Eq. (8). The reference black dashed lines display the power laws $T^{-4}$ and $T^{-2}$. Parameters: $\mathcal{E} = \Theta_x = \Theta_y = \Theta$ and $\tau_{j+1} - \tau_j = T/6$. (b) Adiabatic threshold time $T_{ad}$ (units $\Theta^{-1}$) as a function of $\mathcal{E}$ (units $\Theta$) for fixed $\Theta_y = \Theta_x = \Theta$.

$\Theta_y$, the smallest values of $T_{ad}$ are actually reached for $\mathcal{E}$ of the order of $\mathcal{E} \sim \Theta_y$. This can be understood by observing that, for $\mathcal{E}, \Theta_y < (1+\sqrt{2})\Theta$, the smallest energy gap $\Delta E$ throughout the loop $\Gamma$ is reached for $t = \tau_2$ and it reads

$$\Delta E(\tau_2) = 2\left(\sqrt{\mathcal{E}^2 + \Theta_y^2} - \max\{\mathcal{E}, \Theta_y\}\right). \tag{27}$$

This gap is maximized for $\Theta_y = \mathcal{E}$, which is qualitatively consistent with the fact that $T_{ad}$ is minimized for $\mathcal{E} \sim \Theta_y$. The lack of a quantitative agreement stems from the fact that the energy spectrum of the system [see Fig. 3(b)] features a more complicated structure with respect to simple Landau-Zener transitions. The central role played by $\mathcal{E}$ in determining the spectrum of the system, and therefore its adiabatic threshold, is a peculiarity of the "double tunneling" protocol and has no direct counterpart in the standard single-qubit braiding schemes.

In any case, Fig. 4(b) shows that, for a wide range of parameters, $T_{ad}$ lays between $100\Theta^{-1}$ and $200\Theta^{-1}$. A reasonable estimation for the coupling strengths is $\Theta \sim 10\,\mathrm{GHz}$ [32, 35, 56] which corresponds to a reasonable timescale $T_{ad} \sim 10\,\mathrm{ns}$. Longer durations result in even smaller diabatic errors, which are known to be polynomially suppressed in $T$ [56]. In particular, we find a suppression approximately proportional to $T^{-4}$, as shown by the blue line in the inset of Fig. 4(a). The onset of this specific power-law traces back to the fact that we chose an interpolation function $g(l)$ which features a continuous first derivative [see Eq. (8)]. A different choice could lead to a quadratic suppression [56]: see, for example, the red plot in the inset of Fig. 4(a), which is obtained with the interpolation function $\tilde{g}(l) = l$. Similarly to the standard braiding schemes, depending on the required gate fidelity, the assessment and mitigation of diabatic errors might play an important role in realistic implementations of our entangling scheme [56–61].

## 4.2   Finite tuning accuracy and parasitic couplings

A finite tuning accuracy of parameters $\chi_j(t)$ results in deviations from the loop $\Gamma$ specified in Tab. 1. Importantly, in complete analogy with standard holonomic braiding schemes [8,31], both our $4\gamma$ and double tunneling protocols are robust with respect to errors on the control of $\chi_j(t)$. This is due to two important reasons: (i) The topological nature of the Majorana qubits guarantees the existence of parameters (e.g. the separation lengths) which exponentially suppress the coupling/interactions between the MZMs. As a result, the loops in parameter space can be confined to the three coordinate planes $\chi_j = 0$ ($j = x, y, z$) with exponential accuracy. (ii) On each of these coordinate planes, the Berry curvature associated with the holonomic protocols has no components perpendicular to the plane itself. This statement is proven in App. A.2 and it can be readily seen from Eq. (16) for the $4\gamma$ protocol. Poor control over deviations from $\Gamma$ within each coordinate plane has, therefore, no influence on the resulting unitary operation $\mathcal{U}_\Gamma^{ent}$.

Finally, let us discuss the robustness with respect to possible parasitic couplings. In contrast to the $4\gamma$ protocol, the double tunneling scheme only features three of them. In particular, for $\tau_0 < t < \tau_2$ and $\tau_2 < t < \tau_4$, the only unwanted couplings between the involved Majoranas are the inter-qubit terms $\mathcal{I}_{25}$ and $\mathcal{I}_{22}$, respectively. As for the last stage of the protocol $\tau_4 < t < \tau_6$, the only possible parasitic coupling is the intra-qubit term $P_{25}^B$. For the sake of clarity, these three parasitic couplings are shown with dashed-dotted lines in Fig. 3(a).

Importantly, the presence of those three parasitic coupling is not detrimental for the implementation of the holonomic entangling gate. Let us focus, for simplicity, on the first stage of the protocol $\tau_0 < t < \tau_2$. In this case, the only possible parasitic coupling is $\mathcal{I}_{25}$. As it commutes with both the zero energy operators $R_A$ and $R_B$, the degeneracy of the four ground states is preserved. The same holds for the other two stages of the protocol, i.e. $\tau_2 < t < \tau_4$ and $\tau_4 < t < \tau_6$. Moreover, we numerically verify that the presence of any of the three parasitic couplings does not modify the geometrical phases acquired by the qubits state throughout the holonomic protocol. To this end, we simulate the time evolution of the initial state $|i\rangle$ according to the Hamiltonian

$$H^{pc}(t) = H^{dt}(t) + v_1(t)\mathcal{I}_{25} + v_2(t)\mathcal{I}_{22} + v_3(t)P_{25}^B \tag{28}$$

where the parameters $v_j(t)$ evolve according to

$$v_j(t) = \begin{cases} 0 & t > \tau_{2j+2} \vee t < \tau_{2j} \\ V(t - \tau_{2j}) & \tau_{2j} \leq t < \tau_{2j+1} \\ V(\tau_{2j+2} - t) & \tau_{2j+1} \leq t < \tau_{2j+2} \end{cases} \tag{29}$$

and control the strength of the three parasitic couplings. In the adiabatic regime, we verify that the final state still satisfies $|f(T)\rangle = |f(\infty)\rangle$, independently of the maximum value $V$ acquired by $v_j(t)$. As an example, in Fig. 4(a), we plot the overlap $|\langle f(\infty)|f(T)\rangle|^2$ computed with $V = \Theta/2$ (orange thin line).

## 5   Conclusions

Two-qubit entangling gates, such as the CNOT gate, are essential building blocks for quantum computations. In the realm of topological Majorana qubits, these gates are almost exclusively

considered within the framework of measurement-based topological quantum computation. In this paper, we carefully analyze a complementary approach and propose two holonomic entangling protocols. They represent non-trivial extensions to the two-qubit case of a known technique to implement (topological) single-qubit gates.

At the formal level, the holonomic approach is equivalent to the measurement-based schemes [34]. In contrast to the latter, however, our holonomic protocols do not require the implementations of forced and projective measurements of (joint) Majorana parities. We believe that this might be an advantage, especially since efficient and high-fidelity measurement schemes are still lacking. We fully characterize the protocols and prove the high degree of robustness of the *double tunneling* scheme. Importantly, the latter only requires the ability to tune pair-wise Majorana couplings within qubits characterized by a finite charging energy. Those ingredients are actually featured by several proposed setups for TQC, such as Josephson junction arrays [31, 32], Majorana box qubits [36] and the tetron/hexon schemes [35], pointing to the potential experimental relevance of our proposal.

# Acknowledgements

**Funding information**   We acknowledge support by the DFG (SFB1170 ToCoTronics), the Würzburg - Dresden Cluster of Excellence on Complexity and Topology in Quantum Matter - ct.qmat (EXC2147, project-id 390858490), and the Elitenetzwerk Bayern Graduate School on Topological insulators.

# A    Non-Abelian Berry curvature

As stated in Eq. (16), the non-Abelian Berry curvature associated with our holonomic entanglement protocols reads

$$\vec{F}_{nm} = \nabla_{\vec{\chi}} \times i\langle \psi_{nm}(\vec{\chi})| \nabla_{\vec{\chi}} |\psi_{nm}(\vec{\chi})\rangle \tag{30}$$

Its analytical computation requires the knowledge of the four ground states $|\psi_{nm}(\vec{\chi})\rangle$ as a function of the three parameters $\vec{\chi} = (\chi_x, \chi_y, \chi_z)$. For both the Hamiltonians $H^{4\gamma}(\vec{\chi})$ and $H^{dt}(\vec{\chi})$, the expressions of $|\psi_{nm}(\vec{\chi})\rangle$ can be conveniently computed by taking advantage of the conservation of $P_{34}^A$, $P_{34}^B$, $P_{05}^A$ and of the total parity $\mathcal{P}^A\mathcal{P}^B$. This allows us to bring the Hamiltonians into block-diagonal form and simplify their diagonalization.

## A.1    The $4\gamma$ protocol

In addition to the aforementioned four quantities, the $4\gamma$ Hamiltonian $H^{4\gamma}(\vec{\chi})$ also commutes with the parity of each individual qubit $\mathcal{P}^A$ and $\mathcal{P}^B$. The presence of a total of five independently conserved operators ensure the possibility to express $H^{4\gamma}(\vec{\chi})$ in terms of 32 blocks consisting of $2 \times 2$ square matrices. We then identify the four blocks, which the four eigenstates $|\psi_{nm}(\vec{\chi})\rangle$ belong to. To this end, we recall that

$$P_{05}^A|\psi_{nm}(\vec{\chi})\rangle = \mathcal{P}^A|\psi_{nm}(\vec{\chi})\rangle = \mathcal{P}^B|\psi_{nm}(\vec{\chi})\rangle = +|\psi_{nm}(\vec{\chi})\rangle \tag{31}$$

and exploit Eq. (13). The four $2 \times 2$ blocks reads

$$H_{nm}^{4\gamma}(\vec{\chi}) = \chi_x \sigma_y - (-1)^{m+n} \chi_y \sigma_x - \chi_z \sigma_z, \tag{32}$$

where $\sigma_i$ are Pauli matrices. For the sake of completeness, the bases we have chosen read

$$n = 0; \; m = 0 \rightarrow \left\{ |0_{12}^A 0_{24}^A 0_{05}^A 0_{12}^B 0_{24}^B 0_{05}^B\rangle, |0_{12}^A 0_{24}^A 0_{05}^A 1_{12}^B 0_{24}^B 1_{05}^B\rangle \right\}, \tag{33}$$

$$n = 0; \; m = 1 \rightarrow \left\{ |0_{12}^A 0_{24}^A 0_{05}^A 1_{12}^B 1_{24}^B 0_{05}^B\rangle, |0_{12}^A 0_{24}^A 0_{05}^A 0_{12}^B 1_{24}^B 1_{05}^B\rangle \right\}, \tag{34}$$

$$n = 1; \; m = 0 \rightarrow \left\{ |1_{12}^A 1_{24}^A 0_{05}^A 0_{12}^B 0_{24}^B 0_{05}^B\rangle, |1_{12}^A 1_{24}^A 0_{05}^A 1_{12}^B 0_{24}^B 1_{05}^B\rangle \right\}, \tag{35}$$

$$n = 1; \; m = 1 \rightarrow \left\{ |1_{12}^A 1_{24}^A 0_{05}^A 1_{12}^B 1_{24}^B 0_{05}^B\rangle, |1_{12}^A 1_{24}^A 0_{05}^A 0_{12}^B 1_{24}^B 1_{05}^B\rangle \right\}. \tag{36}$$

In the idle configuration, we have $\chi_x = \chi_y = 0$ and $\chi_z > 0$, which allows us to identify the ground states $|\psi_{nm}(\chi_x = \chi_y = 0)\rangle$. The computation of the Berry curvature is then straightforward and yields Eq. (16). Its isotropy stems from the fact that $H_{nm}^{4\gamma}(\vec{\chi})$ treats the three parameters $\chi_i$ on equal ground.

## A.2 The double tunneling protocol

Since the Hamiltonian $H^{dt}(\vec{\chi})$ does not commute with the individual parity of each qubit, it can be only brought in a block diagonal form consisting of 16 square matrices. In analogy with the previous case, we identify the four $4 \times 4$ blocks, which the four eigenstates $|\psi_{nm}(\vec{\chi})\rangle$ belong to. They read

$$H_{00}^{dt}(\vec{\chi}) = H_{11}^{dt}(\vec{\chi}) = \begin{pmatrix} -\chi_z + 2\epsilon & i\chi_x & i\chi_y & i\chi_y \\ -i\chi_x & \chi_z + 2\epsilon & -i\chi_y & -i\chi_y \\ -i\chi_y & i\chi_y & -\chi_z - 2\epsilon & -i\chi_x \\ -i\chi_y & i\chi_y & i\chi_x & \chi_z - 2\epsilon \end{pmatrix}, \tag{37}$$

$$H_{01}^{dt}(\vec{\chi}) = H_{10}^{dt}(\vec{\chi}) = \begin{pmatrix} -\chi_z + 2\epsilon & i\chi_x & -i\chi_y & i\chi_y \\ -i\chi_x & \chi_z + 2\epsilon & -i\chi_y & i\chi_y \\ i\chi_y & i\chi_y & -\chi_z - 2\epsilon & -i\chi_x \\ -i\chi_y & -i\chi_y & i\chi_x & \chi_z - 2\epsilon \end{pmatrix}. \tag{38}$$

$$\tag{39}$$

The four bases we have chosen are

$$n = 0; \; m = 0 \rightarrow \tag{40}$$
$$\left\{ |1_{12}^A 0_{24}^A 0_{05}^A 1_{12}^B 0_{24}^B 0_{05}^B\rangle, |1_{12}^A 0_{24}^A 0_{05}^A 0_{12}^B 1_{24}^B 1_{05}^B\rangle, |0_{12}^A 0_{24}^A 0_{05}^A 0_{12}^B 0_{24}^B 0_{05}^B\rangle, |0_{12}^A 0_{24}^A 0_{05}^A 1_{12}^B 0_{24}^B 1_{05}^B\rangle \right\},$$

$$n = 0; \; m = 1 \rightarrow \tag{41}$$
$$\left\{ |1_{12}^A 0_{24}^A 0_{05}^A 0_{12}^B 1_{24}^B 0_{05}^B\rangle, |1_{12}^A 0_{24}^A 0_{05}^A 1_{12}^B 1_{24}^B 1_{05}^B\rangle, |0_{12}^A 0_{24}^A 0_{05}^A 1_{12}^B 1_{24}^B 0_{05}^B\rangle, |0_{12}^A 0_{24}^A 0_{05}^A 0_{12}^B 1_{24}^B 1_{05}^B\rangle \right\},$$

$$n = 1; \; m = 0 \rightarrow \tag{42}$$
$$\left\{ |0_{12}^A 1_{24}^A 0_{05}^A 1_{12}^B 0_{24}^B 0_{05}^B\rangle, |0_{12}^A 1_{24}^A 0_{05}^A 0_{12}^B 0_{24}^B 1_{05}^B\rangle, |1_{12}^A 1_{24}^A 0_{05}^A 0_{12}^B 0_{24}^B 0_{05}^B\rangle, |1_{12}^A 1_{24}^A 0_{05}^A 1_{12}^B 0_{24}^B 1_{05}^B\rangle \right\},$$

$$n = 1; \; m = 1 \rightarrow \tag{43}$$
$$\left\{ |0_{12}^A 1_{24}^A 0_{05}^A 0_{12}^B 1_{24}^B 0_{05}^B\rangle, |0_{12}^A 1_{24}^A 0_{05}^A 1_{12}^B 1_{24}^B 1_{05}^B\rangle, |1_{12}^A 1_{24}^A 0_{05}^A 1_{12}^B 1_{24}^B 0_{05}^B\rangle, |1_{12}^A 1_{24}^A 0_{05}^A 0_{12}^B 1_{24}^B 1_{05}^B\rangle \right\}.$$

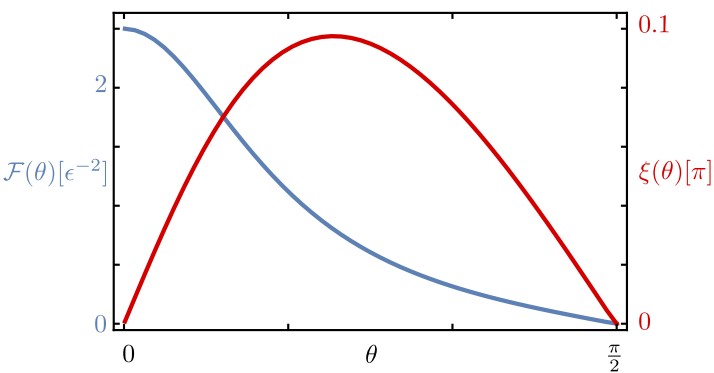

Figure 5: Functions $\mathcal{F}(R, \theta)$ (in blue) and $\xi(R, \theta)$ (in red) for $R = \epsilon$.

Again, the idle configuration allows us to promptly identify the groundstates $|\psi_{00}(\chi_x = \chi_y = 0)\rangle$. The computation of the Berry curvature is lengthy but straightforward. As expected, we obtain $\vec{F}_{00} = \vec{F}_{11} = -\vec{F}_{10} = -\vec{F}_{01}$.

Interestingly, while it is possible to trade $\chi_z$ for $\chi_x$ (and vice versa) with a simple change of basis, the parameter $\chi_y$ plays a different role in the Hamiltonians $H_{nm}^{dt}(\vec{\chi})$. As a result, the Berry curvatures in the parameter space still feature a rotation symmetry around the $\chi_y$ axis but not a full spherical symmetry. This traces back to the peculiar nature of the coupling $\mathcal{O}^{2\gamma+2\gamma}$ we used in the *double tunneling* protocol. Because of the special role played by $\chi_y$, it is convenient to parametrize

$$\vec{\chi} = R(\sin(\theta)\sin(\phi), \cos(\theta), \sin(\theta)\cos(\phi)), \tag{44}$$

which allows us to express the Berry curvature as

$$\vec{F}_{nm}(R, \theta) = \mathcal{F}(R, \theta) \left\{ \cos\left[\xi(R, \theta) + (1 + n + m)\pi\right]\hat{r} + \sin\left[\xi(R, \theta) + (1 + n + m)\pi\right]\hat{e}_\theta \right\} \tag{45}$$

with $\frac{\partial \vec{\chi}}{\partial R} = \hat{r}$ and $\frac{\partial \vec{\chi}}{\partial \theta} = R\hat{e}_\theta$. The modulus of the Berry curvature $\mathcal{F}(R, \theta)$ and the deviations from the radial direction $\xi(R, \theta)$ are plotted in Fig. 5. Importantly, we observe that, on the three coordinate planes $\chi_i = 0$, the Berry connection has no perpendicular components to the planes themselves. This guarantees that the *double tunneling* protocol, despite a more involved structure of the Berry connection, is still topologically protected.

To find the unitary transformation $\mathcal{U}_\Gamma^{ent}$ associated with the holonomic protocol, we compute the non-Abelian Berry phase associated with the $\Gamma$ loop by integrating the Berry curvature on a surface enclosed by $\Gamma$ (or, equivalently, by integrating the Berry connection $\vec{A}_{nm} = i\langle\psi_{nm}(\vec{\chi})|\nabla_{\vec{\chi}}|\psi_{nm}(\vec{\chi})\rangle$ on the loop $\Gamma$).

## B  Robustness with respect to asymmetries in inter-qubit couplings

The time-dependent Hamiltonian $H^{dt}$ considered in Eq. (26) features the same coupling strength $\chi_y(t)$ for both the $\mathcal{I}_{20}$ and $\mathcal{I}_{11}$ inter-qubit couplings. While this assumption greatly simplifies the description of the holonomic entangling protocol, it is unrealistic from the experimental point of view since those coupling will likely be controlled by two independent

| $t$ | $\chi_x(t)$ | $\chi_{y_1}(t)$ | $\chi_{y_2}(t)$ | $\chi_z(t)$ |
|---:|:---:|:---:|:---:|:---:|
| $\tau_0 = 0$ | $0$ | $0$ | $0$ | $\Theta$ |
| $\tau_1$ | $0$ | $0$ | $\Theta_{y_2}$ | $\Theta$ |
| $\tau_2$ | $0$ | $\Theta_{y_1}$ | $\Theta_{y_2}$ | $\Theta$ |
| $\tau_3$ | $0$ | $\Theta_{y_1}$ | $\Theta_{y_2}$ | $0$ |
| $\tau_4$ | $\Theta_x$ | $\Theta_{y_1}$ | $\Theta_{y_2}$ | $0$ |
| $\tau_5$ | $\Theta_x$ | $0$ | $\Theta_{y_2}$ | $0$ |
| $\tau_6$ | $\Theta_x$ | $0$ | $0$ | $0$ |
| $\tau_7$ | $\Theta_x$ | $0$ | $0$ | $\Theta$ |
| $\tau_8 = T$ | $0$ | $0$ | $0$ | $\Theta$ |

Table 2: Values of the coupling strengths $\chi_i(t)$ at the *key* configurations $t = \tau_\alpha$ (with $\alpha = 0, \dots, 8$) along the closed loop in the four-dimensional parameter space.

control knobs. Importantly, however, we show that the entangling protocol is robust with respect to asymmetries in the coupling strengths of $\mathcal{I}_{20}$ and $\mathcal{I}_{11}$. To this end, we focus on the more general time-dependent Hamiltonian

$$H_{As}^{dt}(t) = H_{\mathcal{E}} - \chi_x(t)P_{02}^B - \chi_{y_1}(t)\mathcal{I}_{11} - \chi_{y_2}(t)\mathcal{I}_{20} - \chi_z(t)P_{05}^B \tag{46}$$

and we consider the system to follow a closed loop, in the four-dimensional parameter space, which consists of eight straight lines connecting the eight *key* configurations listed in Tab. 2. Within such a protocol, the coupling strengths $\chi_{y_1}(t)$ and $\chi_{y_2}(t)$ differ in both their maximum values ($\Theta_{y_1}$ and $\Theta_{y_2}$, respectively) and in their time dependence.

In order to characterize the protocol, we firstly observe that the degeneracy of the computational space is preserved. Indeed, analogously to $H^{dt}(t)$, also $H_{An}^{dt}(t)$ commutes with the conserved parity operators, $P_{34}^A$ and $P_{34}^B$, as well as with the operators which transform between the states of the computational space (such as $R_A$ and $R_B$). The spectrum of $H_{An}^{dt}(t)$, which is plotted in Fig. 6(a), confirms that asymmetries in the inter-qubit couplings do not affect the groundstate degeneracy. This allows us to consider the adiabatic evolution of the system along the loop, thus implementing the holonomic entangling process. The latter results in the same unitary operation $\mathcal{U}_\Gamma^{ent}$, detailed in Eq. (18), which characterizes the two other entangling protocols described in the main text. This is testified by Fig. 6(b) where, in complete analogy with Fig. 4(a), we show the overlap between $|f(\infty)\rangle = \mathcal{U}_\Gamma^{ent}|i\rangle$ and the final state $|f(T)\rangle$ obtained after the implementation of the Hamiltonian $H_{As}^{dt}$, with total duration $T$. As the latter increases, the overlap $|\langle f(\infty)|f(T)\rangle^2$ reaches 1. In analogy with the double tunneling protocol, the infidelity is polynomially suppressed in $T$ [see the inset of Fig. 6(b)].

The robustness of the double tunneling protocol with respect to asymmetries in the inter-qubit couplings was to be expected. Indeed, because of the Hamiltonian $\mathcal{E}$, inter-qubit single electron tunnelings are ineffective. The possible unbalances between $\chi_{y_1}$ and $\chi_{y_2}$ have therefore no effect on the entangling process since it is only their contemporaneous presence that matters.

(a)

(b)

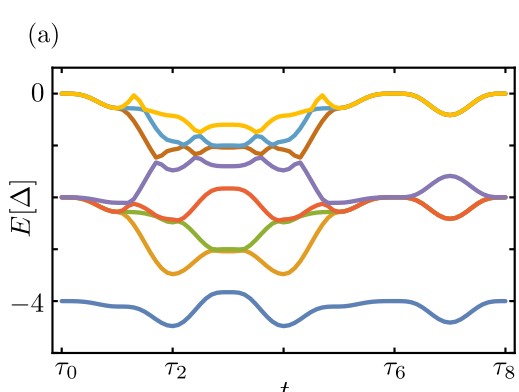
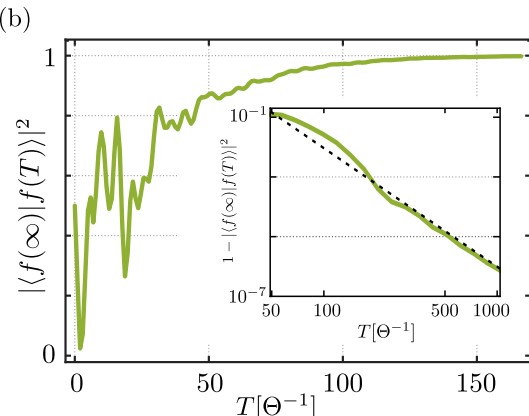

Figure 6: (a) Spectrum of the Hamiltonian $H_{As}^{dt}(t)$ along the closed loop whose *key* configurations are listed in Tab. 2. Only negative energies are shown. Each line is 4-fold degenerate. Energies are in units of $\Theta$. (b) Overlap $|\langle f(\infty)|f(T)\rangle|^2$ as a function of the total duration of the protocol $T$ (in units of $\Theta^{-1}$). The inset display $1 - |\langle f(\infty)|f(T)\rangle|^2$ on a logarithmic scale. The reference black dashed line displays the power law $T^{-4}$. For both panels, we considered the eight *key* configurations to be equally spaced in time, i.e. $\tau_{j+1} - \tau_j = T/8$ and the parameters $\mathcal{E} = \Theta_x = \Theta$, $\Theta_{y_1} = 1.2\Theta$ and $\Theta_{y_2} = 0.8\Theta$.

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
