# Peer review of "Holonomic implementation of CNOT gate on topological Majorana qubits"

_SciPost Physics, doi:SciPost Phys. Core 3, 014 (2020)_

## Round 1 · Referee Report · Anonymous · 2020-9-5

Strengths
* Meets all general acceptance criteria, except one
* Clarity of exposition
Weaknesses
* Lack of novelty (as defined in acceptance criteria)
Report
In this work, the Authors consider the implementation of a two-qubit entangling gate on Majorana-based qubits using the time-dependent manipulation of Majorana couplings, a well known approach for braiding operations.
This work is clearly written, of appropriate length and level of detail, and as far as I can tell, technically valid. My only doubt concerning publication is the degree of novelty that the manuscript offers. Namely, the Authors present two different protocols:
• In the first, they consider a protocol with a tunable joint parity coupling involving four Majoranas. This protocol is essentially the adiabatic evolution version of a projector identity that is used in well-known measurement-based schemes for Majorana entangling gates [see discussion around Eq. 22].
• In the second, the Authors considers two qubits that are charging energy protected, and implement the entangling gate via the simultaneous activation of two tunnel couplings each involving a distinct pair of Majoranas from each qubit. In this configuration an effective four-Majorana coupling (the necessary ingredient to implement a scalable entangling gate) is turned on by co-tunneling processes. This way of obtaining an entangling gate is the adiabatic-evolution version of the four-Majorana measurement discussed e.g. in Ref. 35.
A question on the second protocol. It requires simultaneously tuning two interactions between distinct pairs of Majoranas, which are lumped in a single coupling chi_y(t) in Eq. 26. These two terms are physically distinct and in any implementation will be controlled by independent control knobs (e.g. tunnel gates). It seems unrealistic that these two terms can have an identical strength at all times during the evolution, especially when they need to be turned on. I think that if the two strength differs, this may result in unwanted splitting between computational Majoranas. Have the Authors assessed the impact of this particular imperfection, which seems unavoidable, on fidelity?
Another question is about the estimate for the adiabatic threshold in Fig. 4b. First, I suspect that at the bottom of page 12 the Authors have made a typo when they write that they require the fidelity between final and target state to be smaller than 1e-3. I assume that here the Authors meant that they rather require the infidelity to be 1e-3 or smaller. Under some assumption on energy scales, this requirement translates into an adiabatic gate time of 10 ns or more. This lower bound seems very optimistic, comparable to microwave operations on superconducting qubits. But in any case, a two-qubit gate infidelity of 1e-3 would barely set Majorana qubits apart from the pack of qubits that are much easier to realize. How much would the adiabatic threshold worsen if they had required an infidelity of 1e-6?
To conclude, my doubt regarding publication concerns the degree of novelty of this work, when compared to the acceptance criteria of SciPost Physics. In my opinion this work may better suit SciPost Physics Core instead.
Author: Alessio Calzona on 2020-10-13 [id 1007]
(in reply to Report 2 on 2020-09-05)
We thank the referee for her/his report and for acknowledging the validity and the clarity of our work.
We agree with both referees that the holonomic approach is formally equivalent to measurement-based schemes (as demonstrated in Ref. [34]). We better clarify this point in the new version of the manuscript. At the practical level, however, the implementations of the two approaches differ and pose different experimental challenges (e.g. reaching adiabaticity versus implementing high-fidelity projective measurements). We thus believe that our explicit and detailed analysis of holonomic entangling gates (which has never been done) can be useful for the development of the field which, at least from the experimental point of view, is still in its infancy.
A question on the second protocol. It requires simultaneously tuning two interactions between distinct pairs of Majoranas, which are lumped in a single coupling chi_y(t) in Eq. 26. These two terms are physically distinct and in any implementation will be controlled by independent control knobs (e.g. tunnel gates). It seems unrealistic that these two terms can have an identical strength at all times during the evolution, especially when they need to be turned on. I think that if the two strength differs, this may result in unwanted splitting between computational Majoranas. Have the Authors assessed the impact of this particular imperfection, which seems unavoidable, on fidelity?
We thank the referee for the interesting question. For the sake of simplicity, in the main text, we have grouped two distinct coupling strengths into a single parameter $\chi_y$. As stressed by the referee, in a realistic implementation, the strengths of couplings $I_{11}$ and $I_{20}$ can differ, both in terms of their maximum value and of their time dependence. However, because of the finite charging energy (Hamiltonian H_E), a single inter-qubit tunneling term such as $I_{20}$ (or$I_{11}$) is completely ineffective. Asymmetries between $I_{11}$ and $I_{20}$ are therefore expected not to be detrimental, as long as there is a period in time when both the couplings are on. This picture is confirmed by numerical simulations (which we now provide in the new Appendix B): differences in the two coupling strengths do not split the degeneracy of the computational space and do not modify the Berry phases acquired during the adiabatic evolution. We mention in passing that this very question provides a good example of the importance of our work: our detailed analysis of holonomic entangling protocols allows to directly answer that kind of specific questions, which are definitely relevant for the implementation of the protocols.
Another question is about the estimate for the adiabatic threshold in Fig. 4b. First, I suspect that at the bottom of page 12 the Authors have made a typo when they write that they require the fidelity between final and target state to be smaller than 1e-3. I assume that here the Authors meant that they rather require the infidelity to be 1e-3 or smaller. Under some assumption on energy scales, this requirement translates into an adiabatic gate time of 10 ns or more. This lower bound seems very optimistic, comparable to microwave operations on superconducting qubits. But in any case, a two-qubit gate infidelity of 1e-3 would barely set Majorana qubits apart from the pack of qubits that are much easier to realize. How much would the adiabatic threshold worsen if they had required an infidelity of 1e-6?
We thank the referee for pointing out the typo. Inspired by the comments of both referees, we improved the discussion about the non-adiabatic effects. We now better clarify that the power-law suppression of the non-adiabatic errors (with respect to the total duration of the protocol T) depends on the interpolating function g(l). If the latter features a continuous first derivative, the suppression is proportional to $T^{-4}$. In this case, infidelity of 1e-6 could be reached with an operating time $T \sim 70\div100 \,ns$ (see the blue line in the inset of Fig. 4(a)). By contrast, a linear interpolation function (or the presence of noise) would lead to a weaker suppression, proportional to $T^{-2}$. In this case, infidelities of 1e-3 and 1e-6 would correspond to $T \sim 50\, ns$ and $T \sim\, 1500 ns$, respectively (see the red line in the inset of Fig. 4(a)).
Author: Alessio Calzona on 2020-10-13 [id 1006]
(in reply to Report 3 by Torsten Karzig on 2020-09-11)We thank the referee for his report and for acknowledging the validity and the clarity of our work.
We agree with both referees that the holonomic approach is formally equivalent to measurement-based schemes (as demonstrated in Ref. [34]). We better clarify this point in the new version of the manuscript. At the practical level, however, the implementations of the two approaches differ and pose different experimental challenges (e.g. reaching adiabaticity versus implementing high-fidelity projective measurements). We thus believe that our explicit and detailed analysis of holonomic entangling gates (which has never been done) can be useful for the development of the field which, at least from the experimental point of view, is still in its infancy.
We also thank the referee for his additional comments and suggestions, which allowed us to improve the manuscript in the following ways: - we have corrected the typos; - we better discuss the dependence of the power-law suppression of non-adiabatic errors on the interpolating function g(l); - we have clarified that the spectrum associated with the $4\gamma$ "controlled braiding" protocol is equivalent to the one featured by standard holonomic braiding protocols. Therefore, diabatic effects in the two cases are expected to be completely analogous. We have provided a more comprehensive list of published papers studying non-adiabatic effects in standard braiding protocols.

---

## Round 1 · Referee Report · Torsten Karzig · 2020-9-11

Strengths
1 - Clear presentation. The paper provides a good review of the perspective to interpret topologically protected operations via protected geometric phases.
2- Explicit treatment of non-adiabatic effects in entangling gates.
Weaknesses
1 - Novelty. The main results are essentially known given the equivalence between forced-measurement and adiabatic manipulation (see report).
Report
The authors discuss how to implement entangling gates using an adiabatic evolution rather than measurements. They find that in order to robustly perform the entangling gate a charging energy on the qubit islands is required in order to create the wanted 4 Majorana couplings effectively from virtual cotunneling. The corresponding setup is thus equivalent to the existing proposals for 4 Majorana measurements, with the only difference being that in the evolution-based approach the 4 Majorana term is adiabatically turned on by turning off another (2 Majorana) coupling, while in the measurement version one would first turn off all couplings after an initial 2 Majorana measurement and then turn on the 4 Majorana coupling to perform the measurement.
The outcome of the authors' evolution based scheme is expected since it exactly follows existing measurement-based schemes. The equivalence between adiabatic evolution and forced measurement was explicitly established by Bonderson [Measurement-only topological quantum computation via tunable interactions. PRB 87, 035113 (2013)]. To quote this paper:
"In other words, the effect of time evolution under this adiabatic process [...] is exactly the same as the effect of performing a projective topological charge measurement of the collective charge [...] with predetermined measurement outcome [...] Operationally, this is identical to the “forced measurement” protocol which allows one to effectively perform a topological charge measurement with predetermined measurement outcome. In hindsight, this should perhaps not be so surprising since the adiabatic evolution of ground states includes an implicit continual projection into the instantaneous ground-state subspace and can be thought of as the continuum limit of a series of measurements, with the final measurement being a projection into the final ground-state subspace.
[...] Having established that adiabatic manipulation of interactions can be used to produce a forced measurement operation, it is trivial to use it for anyonic teleportation, braiding, and measurement-only topological quantum computation in precisely the same way as detailed in Refs. 8,9."
In this light, I do not think that the paper meets SciPost's requirements of novelty and impact.
Since the paper studies the effect of (non-)adiabaticity on the evolution-based entangling gate, which hasn't been done explicitly elsewhere, it might be suitable for a publication elsewhere. Below ("requested changes") I provide a list of minor comments/suggestions that the authors may want to consider in another submission.
Requested changes
- page 10, line 5: typo :"is coupled with \gamma^B_2" should use \gamma^B_5 instead
- page 11, line 1: missing reference "[?]"
- discussion of Fig. 4: The authors mention that the time scale depends on the spectrum but it also depends on the form of the time dependence of the ramp. The authors use a function [see Eq. (8)] with a smooth first derivative which gives the 1/T^4 power law suppression of non-adiabatic errors [see Ref. 57]. If they would have used a function with jump in the first derivative or would have considered noise this power law would change to 1/T^2. This would be good to mention.
- Since non-adiabatic effects for the standard braiding sequence have been studied in the literature, it would be useful to know how the standard braid compares to the "controlled braid" considered by the authors. Intuitively, I would guess that there is no qualitative difference.

---

## Editorial Decision

published